# Investigating the Impact of Using IR Bands on Early Fire Smoke Detection from Landsat Imagery with a Lightweight CNN Model

Liang Zhao [1], Jixue Liu [1,*], Stefan Peters [1], Jiuyong Li [1], Simon Oliver [2] and Norman Mueller [2]

1    UniSA STEM, University of South Australia, Mawson Lakes, SA 5095, Australia; liang.zhao@mymail.unisa.edu.au (L.Z.); stefan.peters@unisa.edu.au (S.P.); jiuyong.li@unisa.edu.au (J.L.)
2    Geoscience Australia, 101 Jerrabomberra Ave., Symonston, ACT 2609, Australia; simon.oliver@ga.gov.au (S.O.); norman.mueller@ga.gov.au (N.M.)
*    Correspondence: jixue.liu@unisa.edu.au; Tel.: +61-8-8302-3054

**Abstract:** Smoke plumes are the first things seen from space when wildfires occur. Thus, fire smoke detection is important for early fire detection. Deep Learning (DL) models have been used to detect fire smoke in satellite imagery for fire detection. However, previous DL-based research only considered lower spatial resolution sensors (e.g., Moderate-Resolution Imaging Spectroradiometer (MODIS)) and only used the visible (i.e., red, green, blue (RGB)) bands. To contribute towards solutions for early fire smoke detection, we constructed a six-band imagery dataset from Landsat 5 Thematic Mapper (TM) and Landsat 8 Operational Land Imager (OLI) with a 30-metre spatial resolution. The dataset consists of 1836 images in three classes, namely "Smoke", "Clear", and "Other_aerosol". To prepare for potential on-board-of-small-satellite detection, we designed a lightweight Convolutional Neural Network (CNN) model named "Variant Input Bands for Smoke Detection (VIB_SD)", which achieved competitive accuracy with the state-of-the-art model SAFA, with less than 2% of its number of parameters. We further investigated the impact of using additional Infra-Red (IR) bands on the accuracy of fire smoke detection with VIB_SD by training it with five different band combinations. The results demonstrated that adding the Near-Infra-Red (NIR) band improved prediction accuracy compared with only using the visible bands. Adding both Short-Wave Infra-Red (SWIR) bands can further improve the model performance compared with adding only one SWIR band. The case study showed that the model trained with multispectral bands could effectively detect fire smoke mixed with cloud over small geographic extents.

**Keywords:** remote sensing; multispectral satellite imagery; smoke detection; fire detection; moderate spatial resolution; deep learning

## 1. Introduction

Wildfires can develop quickly, aggravated by climate change, causing substantial consequences to society, ecology, and the economy [1–3]. Fire detection in early stages can prevent the disastrous impact of extreme fires. Using satellite imagery for fire detection is cost-effective since an increasing number of satellites are being launched to monitor the earth. However, detecting early fires from satellite imagery is challenging since the fires can be easily obscured by the thick canopy, clouds, or the smoke they emit. Even when using the thermal band, fires can be masked by the heated background when the weather is hot, and false alarms could be frequently caused by heated bare soils or deserts or other highly reflective regions [4–6].

Detecting fire smoke to infer fires is a better option than direct fire detection, considering fire smoke has the following characteristics: (1) fire smoke can rise above the canopy in a short time and usually has distinctive colours from the vegetation; (2) fire smoke disperses quicker into a large scale than the spread of fire; hence it is easier to be

detected from satellites; (3) the temperature of fire smoke is significantly lower than the hot background.

Nevertheless, fire smoke detection from satellites is challenged by other factors: (1) variant characteristics of fire smoke such as its shape, colour, and scale; (2) similarity and overlap in the spectral signatures between fire smoke and other objects such as snow, cloud, and dust [7–10]. Figure 1 shows the variants of fire smoke in different scenarios captured by Landsat 8 OLI, visualised in true colour using bands 4 (red), 3 (green), and 2 (blue).

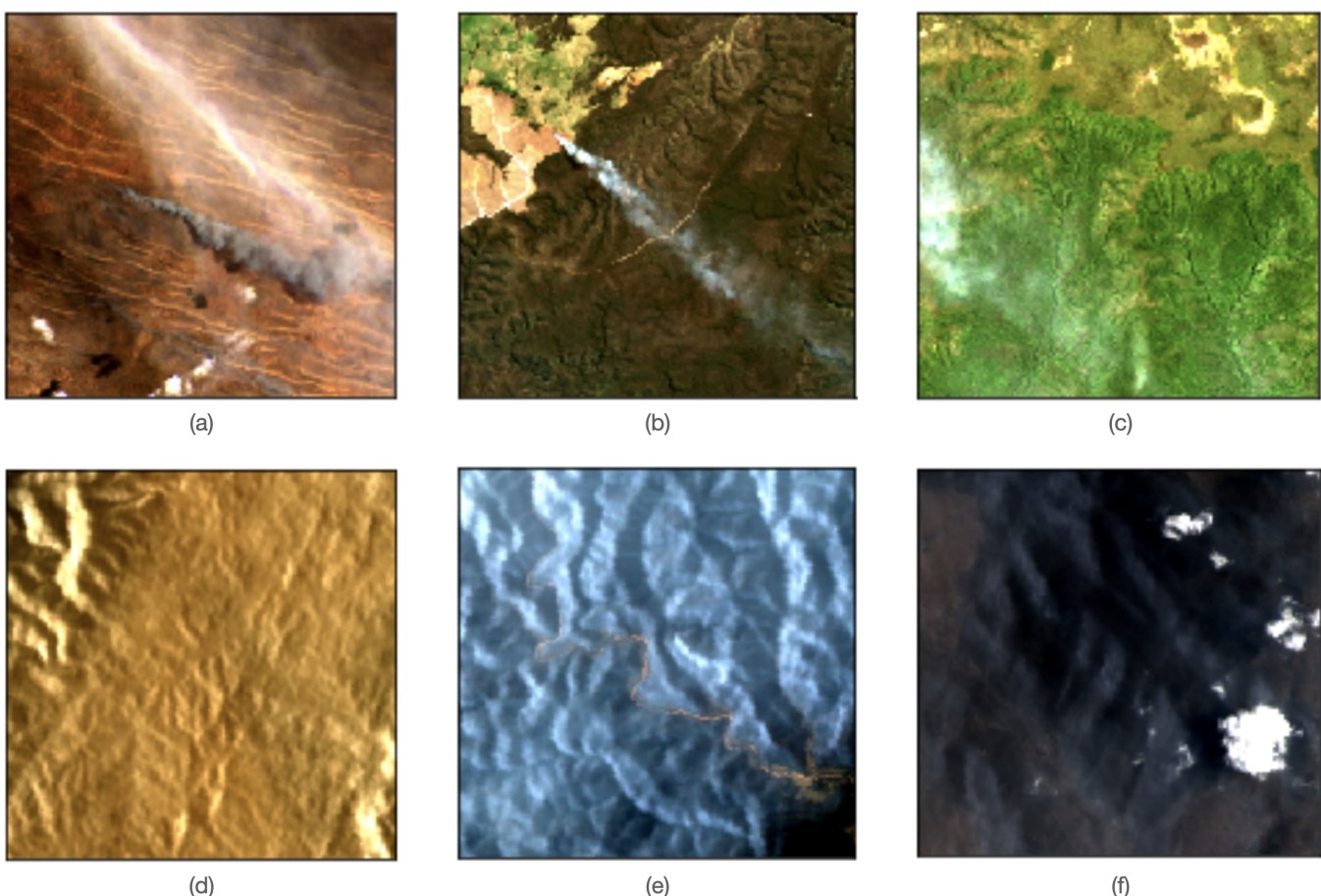

(a)       (b)       (c)

(d)       (e)       (f)

**Figure 1.** Variants of fire smoke in Landsat 8 OLI true-colour imagery. (**a**) Dark grey fire smoke plumes under cirrus clouds. (**b**) Long slim fire smoke plume in bright colour. (**c**) Dispersed fire smoke on the edge of the image. (**d**) Brown-coloured dense fire smoke in the whole image. (**e**) Wide, dispersed fire smoke in light blue colour covering the whole image. (**f**) Spread dense fire smoke in dark grey colour under altocumulus clouds.

Early research tried to discriminate fire smoke in the satellite imagery from other confounding objects (e.g., water, snow, cloud) based on shallow handcrafted features at the pixel level [11–18]. Such features have strong associations with various local conditions and need to be properly redefined in a different area.

The development of Deep Learning (DL) techniques, especially Convolutional Neural Networks (CNN), shifted the research focus in recent years to detecting fire smoke in the satellite imagery at the scene level [10,19]. The DL models can automatically extract deep semantic features to determine whether the satellite imagery contains fire smoke, regardless of the shape, position of the fire smoke, and even when there are other confounding objects or aerosols in the imagery. However, the below gaps are yet to be filled:

- Previous DL-based research was based on satellite imagery with a low spatial resolution (e.g., 0.25–1 km in MODIS imagery and 0.5–2 km in Himawari-8 Advanced Himawari Imager (AHI) imagery), where early fires over small geographic extents could be easily overlooked. Using imagery from satellites with a higher spatial resolution has the advantage of revealing early fires over small geographic extents.
- The existing DL models for fire smoke detection at the scene level only used the visible (referred to as RGB hereinafter) bands of the satellite imagery. Whereas Infra-Red (IR) bands often contain important information that could potentially improve the detection accuracy, particularly if the fires were obscured. For example, in Figure 2, visualising the fire smoke scenes using SWIR_2, NIR, and blue bands reveals the actively burning fires in vivid red colour, burnt scars in dark red colour in the bottom-left image, and fire smoke in light blue colour in both two bottom images. These properties are not clear in the RGB images in the top row.

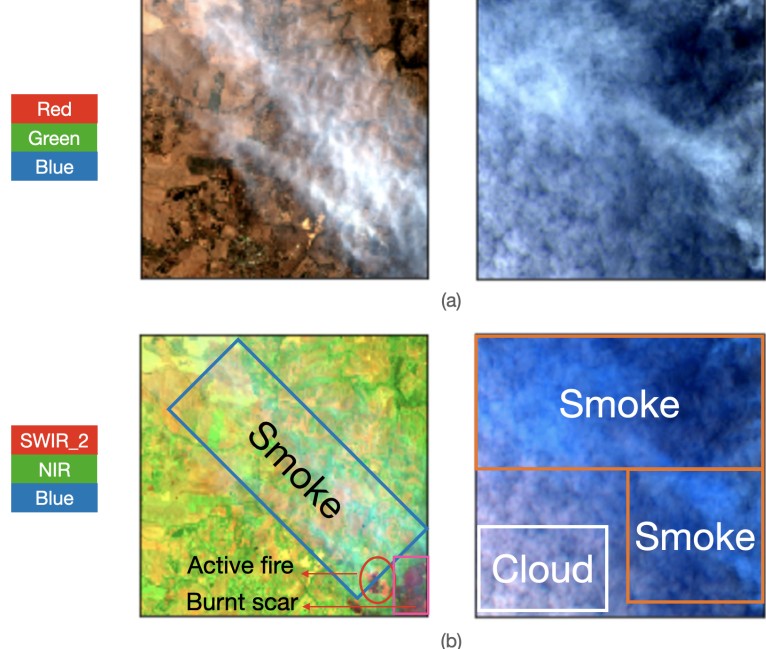

**Figure 2.** Two fire smoke scenes are visualised in different bands. (**a**) RGB. (**b**) SWIR_2, NIR, and blue.

The aims of this work are as follows:

- Construct a labelled multi-class imagery dataset from multispectral moderate spatial resolution satellite imagery and share it with the research community to facilitate the future research for fire smoke detection;
- Investigate the effect of using additional IR bands in DL-based fire smoke detection.

In terms of the former aim, we constructed a fire smoke imagery dataset containing 1836 multispectral images based on Landsat 5 TM and Landsat 8 OLI imagery data. The images contain six spectral bands, including the RGB bands, the Near-Infra-Red (NIR) band, and two Short-Wave Infra-Red (SWIR) bands (i.e., SWIR_1 and SWIR_2), all possessing a 30-m spatial resolution. The details about this dataset will be introduced in Section 3.2.

In terms of the latter aim, we designed a lightweight CNN model allowing a variant number of bands as input for the investigation. We named this model "Variant Input Bands for Smoke Detection (VIB_SD)". The following needs were taken into consideration for designing VIB_SD:

- The latest DL techniques that have been demonstrated effective for fire smoke detection should be integrated to achieve the best possible detection accuracy;

- The model is lightweight in terms of the number of parameters. The lightweight model is time-efficient in training, and resource-efficient for the potential on-board-of-small-satellite applications, which is one of our future research goals.

VIB_SD will be introduced in Section 3.3. Our experiment results demonstrated that adding the NIR band effectively improved the model prediction accuracy, and both SWIR bands can further improve the model prediction accuracy.

In summary, the major contributions of our work presented in this paper are:

- For the first time in the literature, we created a labelled imagery dataset based on Landsat multispectral moderate spatial resolution imagery, which contains three fire smoke scene-related classes (i.e., "Smoke", "Clear", and "Other_aerosol"). This dataset will be expanded to facilitate future research in satellite-based fire smoke detection;
- We designed a lightweight CNN model VIB_SD, which achieved competitive accuracy with the state-of-the-art model SAFA. More importantly, such performance was achieved with less than 2% of the number of parameters used by SAFA. VIB_SD has the potential to be improved and adopted for on-board-of-small-satellite applications.
- Our findings suggest that adding each of the three IR bands (i.e., NIR, SWIR_1, and SWIR_2) individually can effectively improve the fire smoke detection accuracy; while adding all three IR bands collectively can achieve the highest accuracy. The findings provide useful information for the band selection strategies when using multispectral or hyperspectral satellite imagery for fire smoke detection. To the best of our knowledge, such an investigation has not been conducted in the literature.

We will present the remaining content of this paper as follows: Section 2 will review previous related work; Section 3 will introduce the satellite imagery datasets used in this work, the structure of VIB_SD, the experimental settings, and the evaluation metrics; Section 4 will present the experimental results and our findings; Section 5 will demonstrate the effectiveness of using the model trained with multispectral bands in detecting fire smoke mixed with clouds or over small geographic extents; Section 6 will further discuss the results and explore future possibilities; Section 7 will present the conclusion.

## 2. Related Work

### 2.1. Approaches Used in Satellite-Based Fire Smoke Detection

This section will briefly summarise the approaches used in satellite-based fire smoke detection. Table 1 classifies the approaches based on the adopted techniques and the detection levels.

**Table 1.** Approaches used in satellite-based fire smoke detection.

| Type of Approach | Detection Level | Bands Used | Techniques |
|---|---|---|---|
| Non-neural network | Pixel Level | RGB and/or IR | False colour composite |
| | | | Multi-thresholds |
| | | | Traditional machine learning |
| Neural network/DL | Pixel Level | RGB and/or IR | MLP [1], FCN [2] |
| | Scene Level | RGB | CNN |

[1] Multi-Layer Perceptron; [2] Fully Convolutional Network.

Prior to the fast development of DL techniques, the non-neural-network approaches were dominant in the research. False colour composite approaches support the visual exploration of fire smoke. Using different band combinations other than the RGB bands can visually reveal fire smoke in distinctive colours from other objects [20–22]. Such approaches may be used for case analysis but are hardly suitable for automated workflows when working with massive satellite data [7,10].

Multi-threshold approaches tried to discern fire smoke pixels from other confounding pixels using handcrafted threshold values or features based on the reflectance and

brightness temperature values in certain spectral and pseudo bands [11–16]. However, the threshold values and the features are hard to define, as they are strongly associated with the local conditions and solar zenith angles at the time of the image acquisition and vary greatly across different sensor platforms [9,10]. To further detect fire smoke pixels automatically, machine learning techniques, including traditional non-neural network techniques [13,17,18] and Multi-Layer Perceptron (MLP) neural networks [7,9], were employed using training samples extracted from visually classified polygons or by multi-thresholds approaches. Such approaches may have undermined the generalisability due to having few deep semantic features.

Deep convolutional neural networks were more recently used for fire smoke detection at both the pixel level and scene level. A Fully Convolutional Network (FCN) was proposed in [23] to segment smoke pixels from non-smoke pixels. The model was trained using 975 smoke images from Himawari-8 AHI with six spectral bands and one pseudo band (i.e., RGB, NIR, SWIR, top of atmosphere temperature, and fire radioactive power). Although multispectral imagery data were used, this approach is different from the scene level classification that we adopted since the dataset only has one class, "Smoke", at the scene level.

The first scene-level fire smoke DL model SmokeNet was proposed in [10] based on the USTC_SmokeRS dataset, which was constructed by the same authors. Very recently, the state-of-the-art scene-level fire smoke detection model SAFA, trained with the USTC_SmokeRS dataset, was proposed in [19]. SmokeNet and SAFA both used 64% of the USTC_SmokeRS dataset for training, 16% for validation, and 20% for testing. Both models were trained under the same training and testing settings and evaluated with the same metrics. SmokeNet achieved a testing accuracy of 92.75%, compared with 96.22% for the state-of-the-art SAFA.

### 2.2. Techniques Used in DL-Based Scene-Level Fire Smoke Detection

In this section, we will introduce the techniques used in SmokeNet [10] and SAFA [19], which are closely related to the techniques we integrated to design VIB_SD.

Both SmokeNet and SAFA incorporated the attention mechanism [24] to extract salient features. The attention mechanism has been widely used in scene classification models, such as in [25–30]. However, the implementations vary. SmokeNet adopted the channel attention implementation in [31] and implemented the spatial attention module based on a similar algorithm. SAFA implemented its spatial attention module and channel attention module in more complicated ways by incorporating parallel average pooling and max-pooling, feature map transformation with dual kernel size, and learnable coefficients.

Both SmokeNet and SAFA employed residual learning. SmokeNet adopted the residual attention module proposed in [25], which allowed indicative subtle features to be learned for the classification tasks. Instead, SAFA employed the residual blocks in [32] as the backbone blocks and also integrated the residual blocks with the spatial attention and channel attention modules to extract salient features in its Salient Feature Extraction Path (SFEP).

In addition, SAFA proposed a Mutual Activation Interim (MAI) to achieve smooth feature fusion between different levels in its Global Information Extraction Path (GIEP). The prediction from SFEP and GIEP were combined through two learnable coefficients to generate the final prediction.

Attention mechanism and residual learning were both employed in VIB_SD. We adopted the simpler implementations in SmokeNet for spatial and channel attention, considering the lightweight need. Similar to [33], we integrated residual learning in an inception-residual module but with more paths and kernels of different sizes. Instead of adaptively using salient features and global features to improve the classification performance in [19], we combined features extracted in multiple scopes to achieve the same purpose, inspired by the inception structures [33,34]. The structure of VIB_SD and its key modules will be introduced in Section 3.3.

## 3. Materials and Methods

In this section, we will introduce two satellite imagery datasets used in this work, the VIB_SD model and its key modules, the experiment framework, and the evaluation metrics. The software tools used in the Landsat data collection and labelling were developed using Python Jupyter Notebook. We used Tensorflow for the model implementation and training.

### 3.1. RGB USTC_SmokeRS Dataset

The USTC_SmokeRS dataset was used to evaluate VIB_SD by comparing VIB_SD with the existing models trained based on this dataset.

To the best of our knowledge, the USTC_SmokeRS dataset is the only labelled satellite imagery dataset for DL-based scene-level fire smoke detection. The dataset consists of 6225 256 × 256 RGB images collected from MODIS (Level-1B), which has a spatial resolution of 1 km. The dataset contains six fire-smoke-related scene classes, including "Smoke", "Cloud", "Dust", "Haze", "Land", and "Seaside". The number of images in each class of the USTC_SmokeRS dataset is shown in Table 2. More about the dataset can be found in [10].

**Table 2.** Number of images in USTC_SmokeRS.

| Smoke | Cloud | Haze | Dust | Land | Seaside | Total |
|-------|-------|------|------|------|---------|-------|
| 1016 | 1164 | 1002 | 1009 | 1027 | 1007 | 6225 |

### 3.2. Multispectral Landsat Imagery Dataset

One of our contributions in this work is that we constructed a labelled multispectral moderate spatial resolution satellite imagery dataset for early fire smoke detection. The dataset consists of three fire-smoke-related scene classes, namely "Smoke", "Clear", and "Other_aerosol". We used this dataset to investigate the contribution of using additional bands to the fire smoke detection accuracy. We will explain the data collection and labelling processes in this section.

#### 3.2.1. Data Source

We collected the multispectral Landsat imagery data based on historical wildfires in Australia. We note that the Australian-based data does not restrict our methods from being applied in other regions of the world.

We chose the Landsat series as the target satellites since they have a much higher spatial resolution (30 m) than MODIS and Himawari-8 AHI. The major source is Landsat 8 OLI, which was launched on 11 February 2013 and has been in operation since then. To find more images capturing fire smoke successfully, we extended the query time back to 2010, which allowed Landsat 5 TM (decommissioned on 5 June 2013) to be used as a minor part of the data source. Landsat 7 was excluded due to black stripes in its imagery from 31 May 2003 caused by the failure of its scan line corrector. Landsat 9 was already launched on 27 September 2021, but its data were still not publicly available at the time the collection procedures commenced.

We queried and downloaded the surface reflectance Landsat imagery data, which were processed with the algorithm "Nadir Corrected Bi-directional Reflectance Distribution Function Adjusted Reflectance Coupled with a Terrain Illumination Correction (NBART)" [35], from the Digital Earth Australia (DEA) Sandbox platform [36]. The NBART Landsat imagery data were indexed by DEA with open access to the public and can be queried and downloaded based on the range of location coordinates, time, and bands specification.

The time and spatial information needed for querying the data were extracted from historical fire datasets in South Australia (SA) and New South Wales (NSW) hosted on Data SA [37] and Data NSW [38], respectively. Table 3 shows a sample record in the SA historical fires dataset.

**Table 3.** A sample record in the SA historical fires dataset.

| Attributes | Values |
| --- | --- |
| FID | 5801 |
| INCIDENTNU | 202011011 |
| INCIDENTNA | Overland Corner/Calperum |
| INCIDENTTY | Bushfire |
| FIREDATE | 2020-11-15 |
| FINANCIALY | 2020/2021 |
| FIREYEAR | 2020 |
| SEASON | SPRING |
| DATERELIAB | 1 |
| IMAGEINFOR | Landsat 8 17/11/2020 |
| FEATURESOU | 33 |
| CAPTURESOU | 4 |
| HECTARES | 1447.32 |
| SHAPE_Leng | 0.91332 |
| SHAPE_Area | 0.00141367 |
| min_longi | 140.372 |
| max_longi | 140.613 |
| min_lati | −33.7796 |
| max_lati | −33.6953 |
| Sensor | MODIS |
| geometry | (POLYGON ((140.4123221680001 −33.6969758549999…))) |

### 3.2.2. Data Collection Strategy and Processes

It involved two phases to construct the dataset: Imagery collection and tiling and labelling.

1.  Imagery Collection;
    In this phase, the algorithms were developed to extract the time and location information from the historical fire datasets from Data SA and Data NSW and to download Landsat 5 TM and Landsat 8 OLI imagery in a bulk manner based on the derived time and spatial information. We extended the data query time range to cover 16 days (two revisits of the Landsat series) before and after the recorded fire date, which does not strictly indicate the ignition date due to recording discrepancies. This also allowed imagery for the same area to be collected at different times under different weather. We also added a buffering area in the query with 5 km along both the longitude and the latitude, based on the polygon coordinates of the burnt scars. The imagery files returned from the query were visually examined to select those that successfully captured fire smoke.
    We reserved six spectral bands in the imagery data, including the RGB bands, NIR band, SWIR_1 band, and SWIR_2 band. As shown in Table 4, the wavelengths of the six selected bands of Landsat 5 TM and Landsat 8 OLI vary slightly, though each corresponding band falls roughly in the same range. The majority of the imagery data were collected from Landsat 8 OLI. The thermal band was not included because we initially considered constructing a mixed imagery dataset from Landsat and Sentinel-2 (A and B), which can be used to train a model potentially adaptive to different sensors. Since Sentinel-2 does not have a thermal band, it would be better to exclude the thermal band from the Landsat imagery. However, we will consider adding the thermal band in future data collection where applicable.

**Table 4.** Wavelengths of the selected bands of Landsat 5 TM and Landsat 8 OLI.

| Band | Wavelength (µm) | |
| --- | --- | --- |
| | **Landsat 5 TM** | **Landsat 8 OLI** |
| Red | 0.63–0.69 | 0.64–0.67 |
| Green | 0.52–0.60 | 0.53–0.59 |
| Blue | 0.45–0.52 | 0.45–0.51 |
| NIR | 0.76–0.90 | 0.85–0.88 |
| SWIR_1 | 1.55–1.75 | 1.57–1.65 |
| SWIR_2 | 2.08–2.35 | 2.11–2.29 |

From DEA Sandbox, we downloaded 477 imagery files covering fire sites with a wide range of locations, of which eight imagery files were from SA and the rest from NSW. There were 15 imagery files from Landsat 5 TM and the rest from Landsat 8 OLI. The areas covered by the imagery files varied significantly, subject to the scale of the fires.

2.   Tiling and Labelling.
In this phase, we tiled the imagery files to 256 × 256 patches with a 50% overlap rate both horizontally and vertically. The overlap between the patches will help the model learn to recognise fire smoke regardless of the position of the fire smoke in the patches. The tiling process is demonstrated in Figure 3.

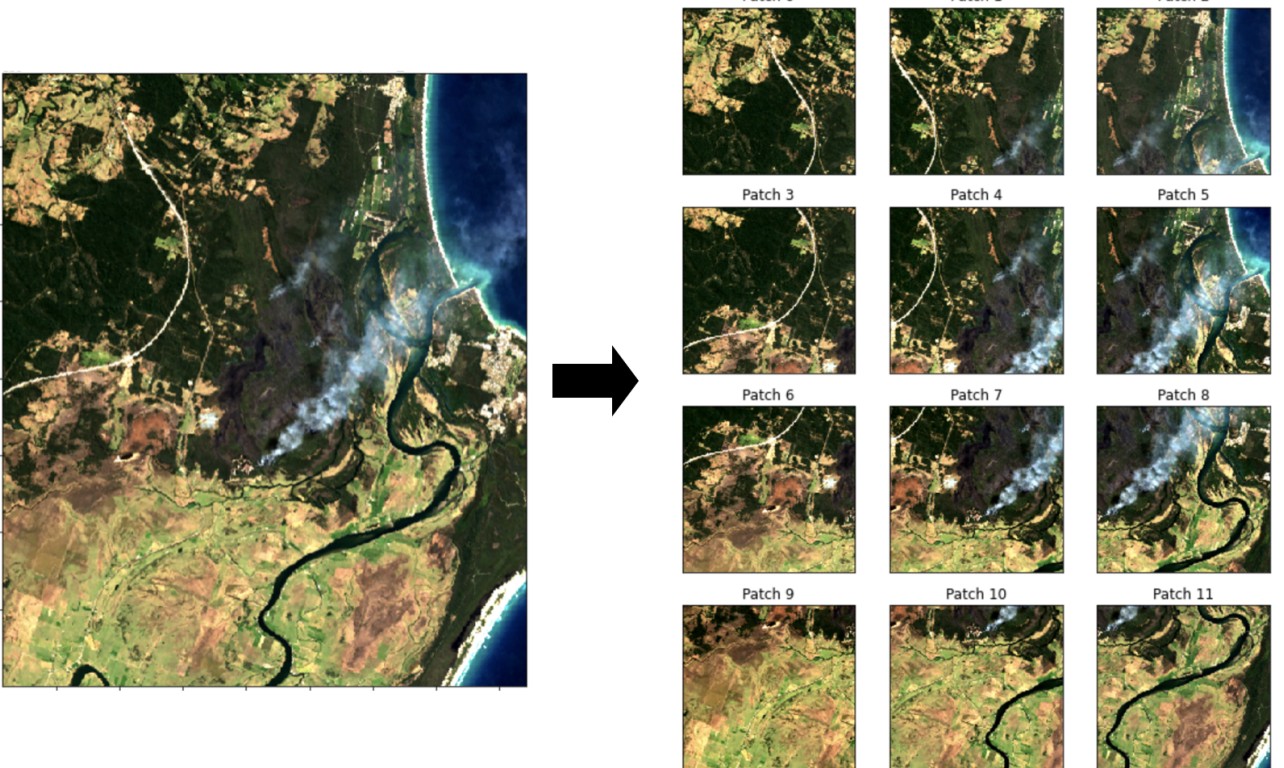

**Figure 3.** Overlapped tiling.

We labelled the patches into three classes: "Smoke", "Clear", and "Other_aerosol". "Other_aerosol" refers to non-smoke scenes that are not "Clear" either, such as scenes with cloud, dust, haze, or other aerosol mixtures. Patches were labelled as "Clear" if there is no visible aerosol or labelled as "Smoke" as long as they contain fire smoke. Identifying fire smoke in the patches is not always easy, as shown in the bottom images in Figure 2. To identify fire smoke more precisely for the labelling, we visually examined the patches in false colour using the SWIR_2 band, the NIR band, and

the blue band. Additional imagery files containing either only a clear background or clouds were downloaded in this phase to balance the number of images in the non-smoke classes.

After tiling and labelling 36 imagery files covering different fire sites, we obtained a training dataset of 1836 256 × 256 images in total, with more than 600 images in each class. The dataset covers a wide range of fire smoke scenes (e.g., fire smoke in different shapes, scopes, colours, and density; fire smoke above different backgrounds; fire smoke mixed with different types of clouds), which broadly reflects the complexity of wildfire events and the challenges that the detection task faces. The experimental results of this paper in Section 4 were obtained using this dataset.

### 3.3. VIB_SD

As mentioned in Section 1, the following needs were considered when designing VIB_SD:

- Using input imagery with multiple combinations of multiple spectral bands.
- Achieve the best possible detection accuracy;
- Lightweight in terms of parameters in the model for high efficiency and potential on-board-of-small-satellite applications.

To achieve good accuracy, we integrated the attention mechanism, residual learning, and the inception structure to assist the extraction of features that are related to fire smoke. We tried to reduce module stacking to control the weight of the model.

Figure 4 shows the main structure of VIB_SD on the left and the structures of the stem block, reduction block, and the classification head on the right.

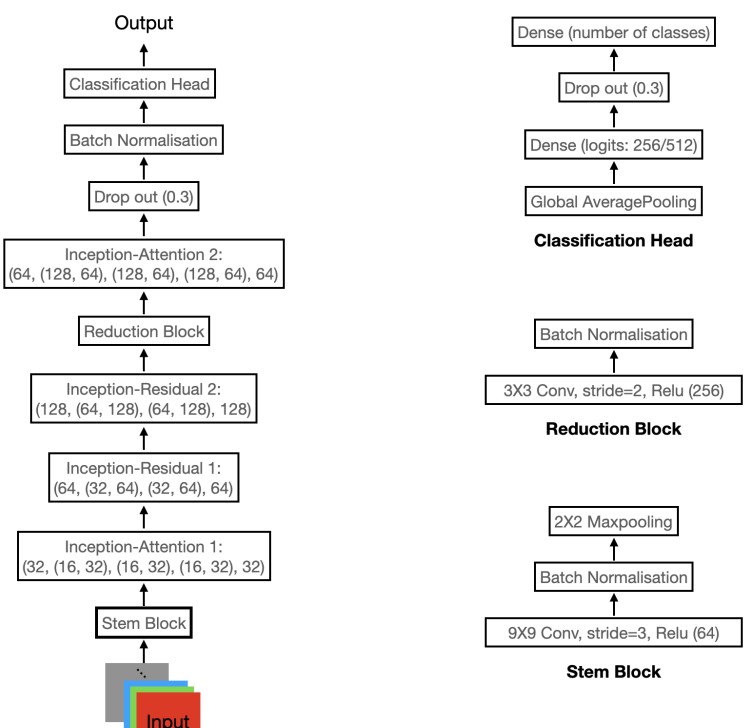

**Figure 4.** Structure of VIB_SD.

The implementations of the key modules in VIB_SD will be introduced in Section 3.3.1. To estimate the performance of VIB_SD, we trained the model using the USTC_SmokeRS dataset and compared the results with those of SmokeNet, SAFA, and Inception-ResNet-V2 [33] under the same training settings. The comparison will be shown in Section 3.3.2. As indicated in Figure 4, the number of logits in VIB_SD was set to 512 when trained with the

USTC_SmokeRS dataset since the dataset has six classes. This number was set to 256 when training VIB_SD with the Landsat dataset as it only has three classes.

The loss function we used for the back propagation was sparse categorical cross-entropy, as defined by the following formula:

$$\text{Loss} = -\sum_{i=1}^{C} y_i \cdot \log \hat{y}_i, \tag{1}$$

where $C$ is the number of classes, $\hat{y}_i$ is the predicted probability of an instance being the $i$-th class and $y_i$ is either 1 if the ground truth label is the $i$-th class or 0 if not.

### 3.3.1. Key Modules

1. Spatial Attention Module;

   The spatial attention module aims to learn the weight for each pixel in each channel of a feature map. The weights are learnt simultaneously. The differences in the weights will help to infer the spatial associations of the pixels, which further helps the model to make the prediction. The module is illustrated in Figure 5a: an input feature map $F = [f_1, f_2, \ldots, f_c] \in \mathbb{R}^{W \times H \times C}$ is first reshaped to a 2D vector $V = [v_1, v_2, \ldots, v_l]$, where $l = W \times H$; $v_i = [p_1^i, p_2^i, \ldots, p_C^i]$ is a 1D vector representing the values of the pixel at position $i$ across all channels in $F$ after $F$ is flattened, and $p_j^i$ is the value of the pixel at the $j$th channel; $V$ is then passed to two fully connected layers both activated by a sigmoid function; the dimension of the interim output was reduced by a ratio $r = 16$ [10] to achieve less computing complexity; the output is then reshaped to generate the spatial attention distribution $S = [s_1, s_2, \ldots, s_C]$, where $s_j \in \mathbb{R}^{W \times H}$ is the spatial attention distribution of $f_j$; the final output of the spatial attention module $O^s = [o_1^s, o_2^s, \ldots, o_C^s]$ is obtained by multiplying the spatial attention distribution $S$ to $F$, where $o_j^s = s_j \times f_j$. Readers can refer to [10] for more details.

2. Channel Attention Module;

   The channel attention module aims to learn the weight of each channel in a feature map. This weight indicates the importance of the channel in predicting the class of the image. The module is illustrated in Figure 5b: for any input feature map $F = [f_1, f_2, \ldots, f_c] \in \mathbb{R}^{W \times H \times C}$, a global average pooling is firstly operated to generate a vector $A = [a_1, a_2, \ldots, a_C]$, where $a_j \in \mathbb{R}$; $A$ is then transformed using two fully connected layers with a dimension reduction ratio $r = 16$ [10], activated by a Relu function and a sigmoid function, respectively; the transformed output is the channel attention distribution $C = [c_1, c_2, \ldots, c_C]$, where $c_j \in \mathbb{R}$ is the weight of channel $f_j$; the final output of the channel attention module $O^c = [o_1^c, o_2^c, \ldots, o_C^c]$ is then obtained by multiplying the channel attention distribution $C$ to $F$, where $o_j^c = c_j \times f_j$. Readers can refer to [10,31] for more details.

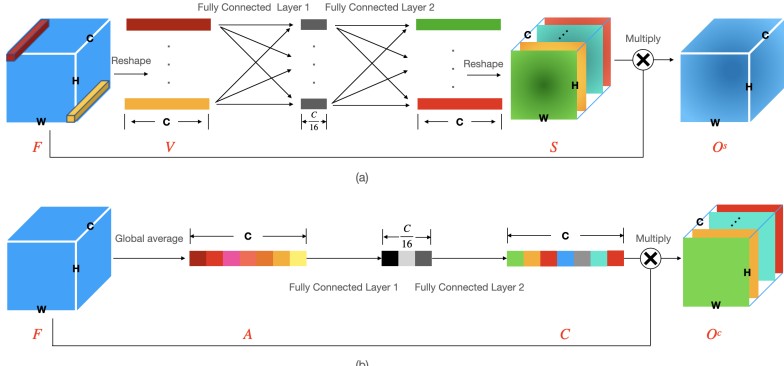

**Figure 5.** Attention modules. (**a**) Spatial attention. (**b**) Channel attention.

3.  Inception-Residual Module;

    The inception-residual module aims to learn residuals associated with spatial features in various scopes since information in the residuals may be important for detecting early fire smoke that usually presents in a small area in the image. We used a four-path inception block with kernels of different sizes to achieve this purpose, which is different from the inception-residual block in [33]. The inception-residual module in VIB_SD is illustrated in Figure 6.

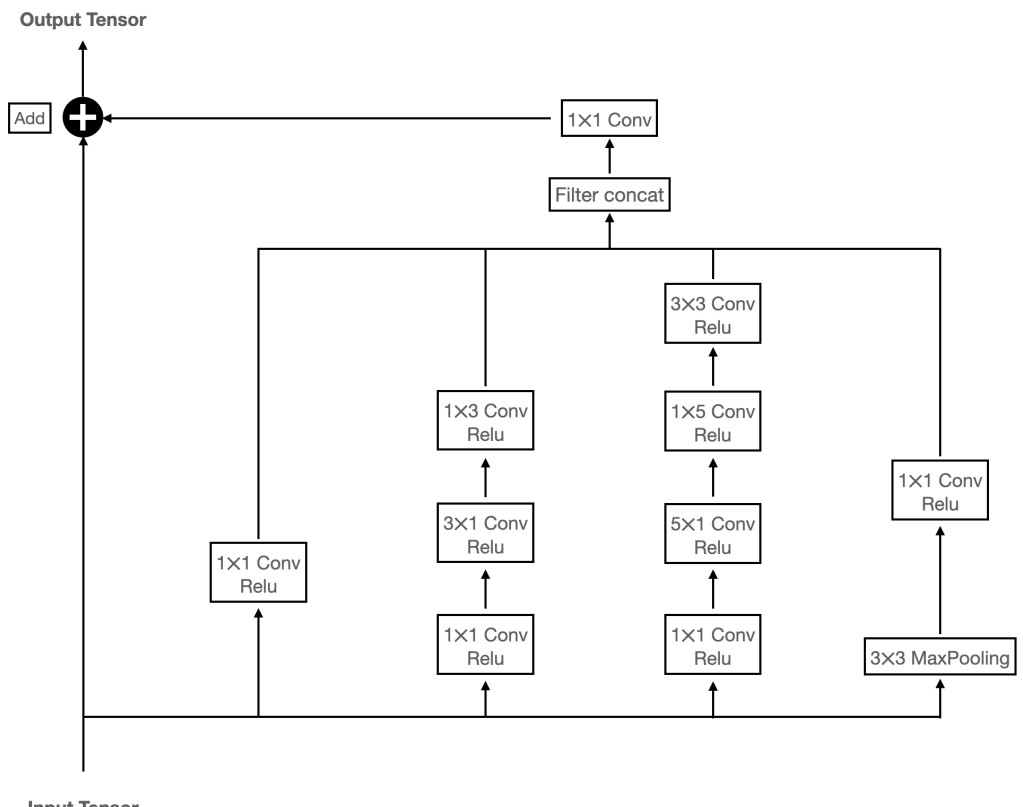

**Figure 6.** Inception-residual module.

4.  Inception-Attention Module.

    The inception-attention module in VIB_SD aims to apply the attention mechanism to spatial features in various scopes extracted using kernels of different sizes. Specifically, we try to extract spatial features in three different scopes through three paths. In the first path, we use a $3 \times 1$ kernel followed by a $1 \times 3$ kernel to extract spatial features in small scopes. In the second path, we use a $7 \times 1$ kernel followed by a $1 \times 7$ kernel to extract spatial features in medium scopes. In the third path, we use an $11 \times 1$ kernel followed by a $1 \times 11$ kernel to extract spatial features in large scopes. The three paths are each followed by a spatial attention module. We use the other two paths to generate feature maps containing less spatial information. One path only uses a $1 \times 1$ kernel; the other uses a $3 \times 3$ max-pooling layer followed by a $1 \times 1$ kernel. The feature maps generated from the five paths are concatenated, after which we use a channel attention module to allocate weights to the channels in the new feature map. This will help the model predictions based on the importance of the extracted spatial features. The inception-attention module in VIB_SD is demonstrated in Figure 7.

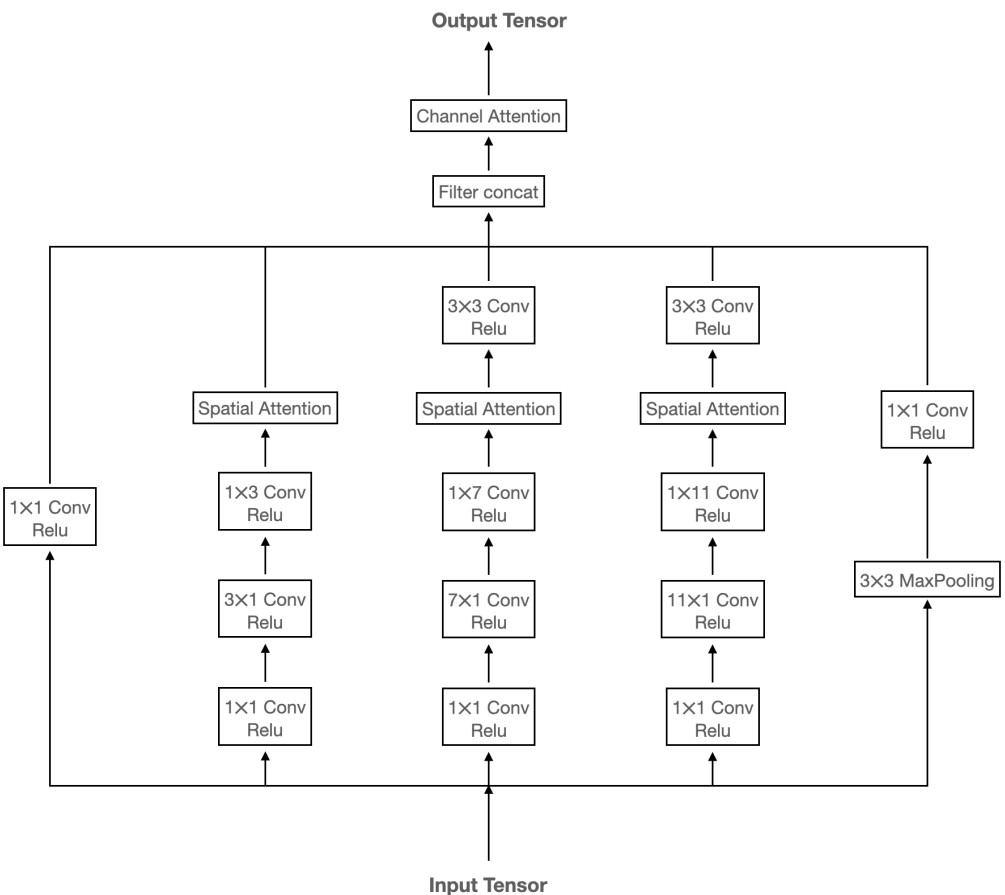

**Figure 7.** Inception-attention module.

### 3.3.2. Comparison with the State-of-the-Art

In this section, we will demonstrate the proposed lightweight model VIB_SD can achieve competitive performance with the state-of-the-art models, even though it has much fewer parameters.

We compared VIB_SD with SmokeNet, SAFA, and Inception-ResNet-V2 [33]. SmokeNet was the best model prior to SAFA, which is the state-of-the-art. Inception-ResNet-V2 contains a similar key component to VIB_SD: the inception-residual block. Many other scene level classification DL models could be also used for fire smoke detection, particularly those developed for remote sensing applications, for example, BoCF [39], RSSC-ETDL [40], LPDCMEN [41], D-CNN [42], and KFBNet [43]. However, these models will not be explained provided they have been compared with SmokeNet and SAFA in [10,19], and comparing models is not the aim of this work.

Since both SmokeNet and SAFA were trained using the USTC_SmokeRS dataset, we trained VIB_SD and Inception-ResNet-V2 using the same dataset for the comparison.

We split the dataset for training (64%), validation (16%), and testing (20%) and set the batch size as 32, the same as in [10,19].

We followed the same data pre-process: The input images for the training were resized to $230 \times 230$ before randomly cropping to $224 \times 224$. Then, we augmented the training images with random horizontal and vertical flipping. The input images for validation and testing were resized to $224 \times 224$ directly. All the images were standardised using the "per_image_standardization" function provided in Tensorflow.

We used Adam for optimisation and dynamically reduced the learning rate from 0.01 by a factor of 0.2 when the validation loss failed to decrease after 20 epochs. We increased our max epochs to 500 compared with 200 in [10,19] since we noticed the training accuracy still has space to improve after 200 epochs. We applied early stopping when the validation

accuracy failed to increase within 90 epochs, which aims to avoid redundant training while guaranteeing the training performance.

We used accuracy and kappa-coefficient as the evaluation metrics since they were adopted in [10,19]. The definition of accuracy and kappa-coefficient will be introduced in Section 3.5.

Table 5 compares the number of parameters, accuracy, and kappa-coefficient of the four models. The results show that VIB_SD significantly reduced the parameter number with minor compromises in accuracy compared with the state-of-the-art model SAFA. However, VIB_SD achieved higher accuracy and kappa-coefficient than both SmokeNet and Inception-ResNet-V2.

**Table 5.** Model performance comparison.

| Model | Parameters | Accuracy | Kappa-Coefficient |
|---|---|---|---|
| SmokeNet | 53.5 M | 92.75% | 0.9130 |
| SAFA | 84.2 M | 96.22% | 0.9546 |
| Inception-ResNet-V2 | 54.4 M | 91.33% | 0.8958 |
| VIB_SD | 1.66 M | 93.57% | 0.9227 |

### 3.4. Experiments

The objectives of the experiments are to:

- Demonstrate the effectiveness of using additional IR bands in accurate fire smoke detection;
- Examine the contributions of NIR and SWIR bands to the model prediction accuracy.

We trained the VIB_SD model using five different band combinations. The band combinations are RGB, RGBN, RGBNS1, RGBNS2, and RGBNS1S2, where N refers to the NIR band, S1 refers to the SWIR_1 band, and S2 refers to the SWIR_2 band. The trained models are named VIB_SD_RGB, VIB_SD_RGBN, VIB_SD_RGBNS1, VIB_SD_RGBNS2, and VIB_SD_RGBNS1N2 accordingly.

With the five models, the contribution of different bands can be verified as follows: The contribution of NIR can be verified by comparing VIB_SD_RGBN to VIB_SD_RGB; the contribution of the SWIR bands can be verified by comparing VIB_SD_RGBNS1 or VIB_SD_RGBNS2 to VIB_SD_RGBN; the individual contribution of each SWIR band can be verified by comparing VIB_SD_RGBNS1S2 to VIB_SD_RGBNS1 and VIB_SD_RGBNS2.

Experimental Settings

The models were trained using the multispectral moderate spatial resolution Landsat imagery dataset we collected. We used 64% of the dataset for training, 16% for validation, and 20% for testing. The number of images in each class is shown in Table 6.

**Table 6.** Components of the Landsat dataset.

| Smoke | Other_aerosol | Clear | Total |
|---|---|---|---|
| 615 | 605 | 616 | 1836 |

All images in the training data were augmented with random horizontal and vertical flipping. We did not apply standardisation since we achieved better training performance without doing so. Augmentation was not applied to the images for validation and testing.

The loss function, regularisation, and optimisation are the same as the settings described in Section 3.3.2.

We compared the five models in two ways:

- Each model was trained 10 times with random samples following the above split ratios. The samples obtained may be different for each split. The overall performance of the models was compared;

- All the five models were trained using the same training samples and testing samples in one random split. The performance of the models was compared.

The former trained the models with more variations of data, while the latter made the models more comparable.

### 3.5. Evaluation Metrics

As mentioned in Section 3.3.2, we adopted accuracy and kappa-coefficient as the evaluation metrics.

The formulas for calculating the accuracy and the kappa-coefficient are defined in Table 7. $N$ denotes the total number of images; $i$ refers to a specific class; $N_{ii}$ is the number of true positive predictions of class $i$; $N_{i+}$ denotes the number of images of class $i$ that were classified as other classes; $N_{+i}$ denotes the number of images of other classes that were classified as class $i$.

**Table 7.** Formulas of accuracy and kappa-coefficient.

|  | **Predicted Class 1** | $\cdots$ | **Predicted Class t** |
|---|---|---|---|
| **Actual Class 1** | $N_{11}$ | $\cdots$ | $N_{1t}$ |
| $\vdots$ | $\vdots$ | $\cdots$ | $\vdots$ |
| **Actual Class t** | $N_{t1}$ | $\cdots$ | $N_{tt}$ |
| **Accuracy** | $\dfrac{\sum_1^t N_{ii}}{N}$ | | |
| **Kappa-coefficient** | $\dfrac{N \sum_1^t N_{ii} - \sum_1^t (N_{i+} N_{+i})}{N^2 - \sum_1^t (N_{i+} N_{+i})}$ | | |

### 4. Results

Based on the 10 results of each model, we obtained their accuracy range within the 95% confidence interval of the mean value, best accuracy, the kappa-coefficient range within the 95% confidence interval of the mean value, and best kappa-coefficient, as listed in Table 8, which also includes the number of parameters of the models.

**Table 8.** Performance of models using variant bands based on 10 results.

|  | **VIB_SD_RGB** | **VIB_SD_RGBN** | **VIB_SD_RGBNS1** | **VIB_SD_RGBNS2** | **VIB_SD_RGBNS1S2** |
|---|---|---|---|---|---|
| **Parameters** | 1.660 M | 1.666 M | 1.671 M | 1.671 M | 1.68 M |
| **Accuracy** | $83.28 \pm 1.57\%$ | $87.78 \pm 1.38\%$ | $87.78 \pm 1.12\%$ | $86.4 \pm 0.09\%$ | $86.21 \pm 1.18\%$ |
| **Best-Accuracy** | 86.45% | 92.41% | 89.97% | 89.43% | 89.16% |
| **Kappa** | $0.7488 \pm 0.0234$ | $0.8164 \pm 0.0207$ | $0.8164 \pm 0.0168$ | $0.7956 \pm 0.0135$ | $0.7929 \pm 0.0178$ |
| **Best-Kappa** | 0.7964 | 0.8861 | 0.8491 | 0.8413 | 0.8373 |

Based on Table 8, VIB_SD_RGB has the worst accuracy and kappa-coefficient, while VIB_SD_RGBN has the best accuracy and kappa-coefficient under both criteria. This implies that adding the NIR band can improve the model performance. However, the accuracy and kappa-coefficient decreased unexpectedly when the SWIR bands were added on top of the NIR band. Particularly, the accuracy and kappa-coefficient saw a larger decrease when both SWIR bands were added compared to when only one SWIR band was added. Potential reasons that might be associated with such results will be discussed in Section 6.

Figure 8 shows the boxplots of the accuracy and kappa-coefficient from the 10 results of the five models.

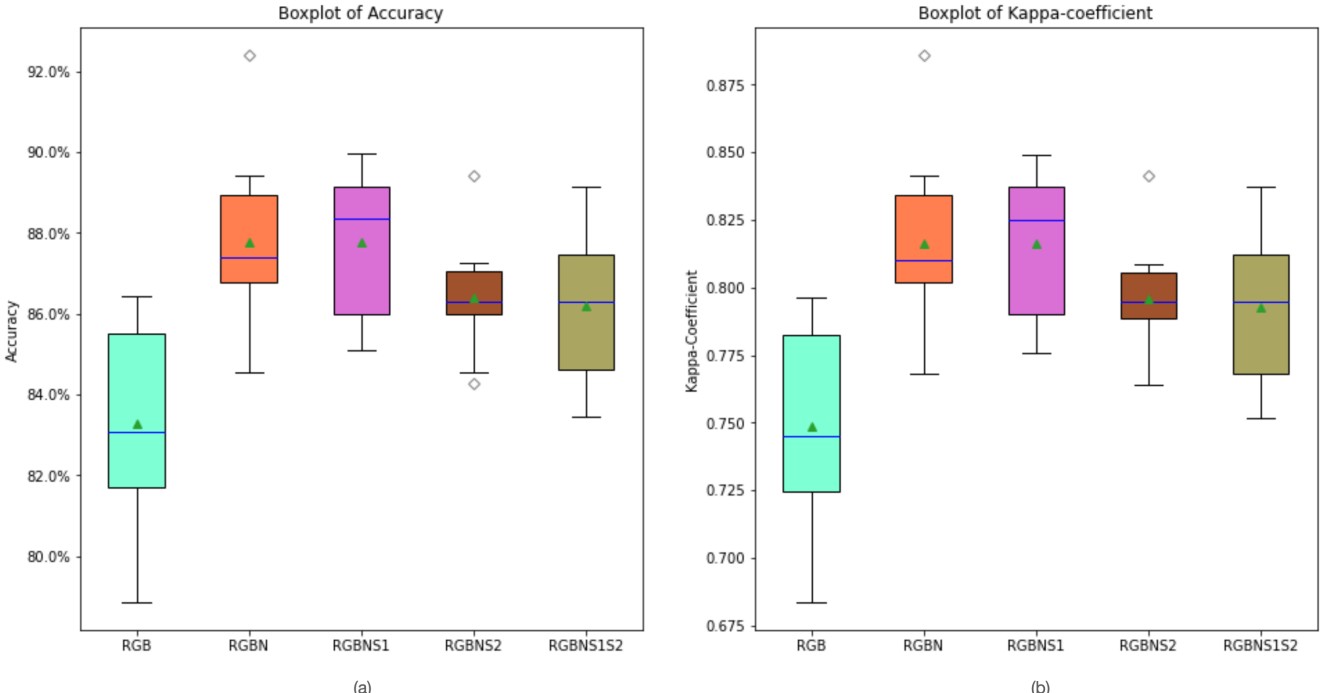

**Figure 8.** Boxplots based on the 10 results. (**a**) Accuracy. (**b**) Kappa-coefficient.

The boxplots in Figure 8 indicate that VIB_SD_RGB is less effective than the other four models. In contrast to what Table 8 implies by evaluating the models with mean values, Figure 8 suggests that VIB_SD_RGBNS1 has the highest median accuracy and kappa-coefficient.

Table 9 shows the results of the five models trained and tested using the same samples obtained in one split. Since all models were trained with the same training samples and tested with the same testing samples, the models can be compared more fairly, although the results may not represent the best possible accuracy and kappa-coefficient of the models. We used boldface font to indicate the best accuracy and kappa-coefficient.

**Table 9.** Performance of models using variant bands with the same samples.

| Model | Testing Accuracy | Kappa-Coefficient |
|---|---|---|
| VIB_SD_RGB | 83.20% | 0.7483 |
| VIB_SD_RGBN | 84.82% | 0.7723 |
| VIB_SD_RGBNS1 | 85.64% | 0.7842 |
| VIB_SD_RGBNS2 | 85.64% | 0.7843 |
| VIB_SD_RGBNS1S2 | **86.45%** | **0.7966** |

Table 9 presents different results: the more bands a model has, the better performance it can achieve. It is also worth noting that VIB_SD_RGBNS1 and VIB_SD_RGBNS2 achieved the same prediction accuracy with slightly different kappa-coefficient. This means although the two models both correctly predicted the same number of images, the true positive predictions in each class varied.

Tables 8 and 9 and Figure 8 all imply that using additional IR bands can effectively increase the model prediction accuracy. Particularly, adding the NIR band greatly improved the prediction accuracy compared with only using the RGB bands.

Based on the fair comparison results in Table 9, it can be inferred that:

- Both the SWIR_1 and SWIR_2 bands contain useful information for fire smoke detection; adding either one of them has a similar contribution to the improvement of the prediction accuracy;
- The SWIR_1 and SWIR_2 bands contain distinctive information; adding both can further improve the prediction accuracy.

## 5. Case Study

To examine the effectiveness of using multispectral moderate spatial resolution imagery for fire smoke detection, we used VIB_SD_RGBNS1S2 to conduct predictions on four different fire smoke scenes captured by Landsat 8 OLI, which are shown in Figure 9. All four scenes have not been used to generate the training dataset. We selected the best weights of VIB_SD_RGBNS1S2, which yielded the highest accuracy of 89.16% for the predictions.

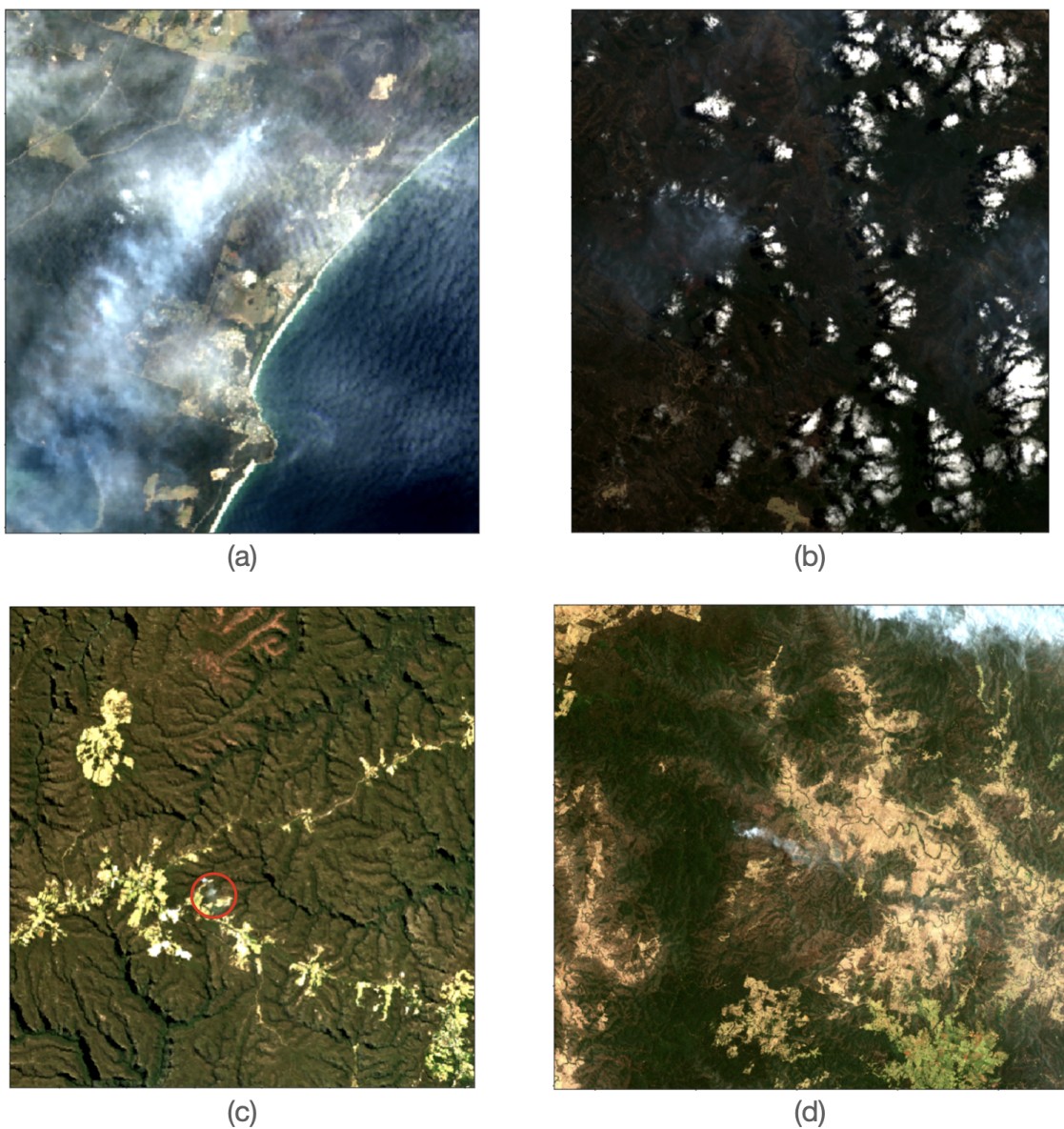

**Figure 9.** Fire smoke scenes. (**a**) Fire smoke mixed with thin clouds above the seaside. (**b**) Diffused fire smoke at multiple sites under altocumulus clouds. (**c**) Cloud-free fire smoke (in the red circle) over a very small geographic extent. (**d**) Cloud-free fire smoke plumes in different scales at two different sites.

The fire smoke scenes were tiled first, the prediction was then conducted on the patches. Since the area covered in scene (a) is too small to be properly tiled with a 50% overlap, we increased the overlap rate to 75% for scene (a) when conducting the prediction. The overlap rate in the prediction for the other three scenes remained at 50%.

The prediction results of scene (a) are shown in Figure 10. In the prediction results, the text above each patch shows the id of the patch, the predicted class (CLR refers to "Clear", SMK refers to "Smoke", OA refers to "Other_aerosol"), and the probability of the predicted class; the text under each patch shows the probabilities of the patch being "Clear", "Other_aerosol", or "Smoke" from left to right.

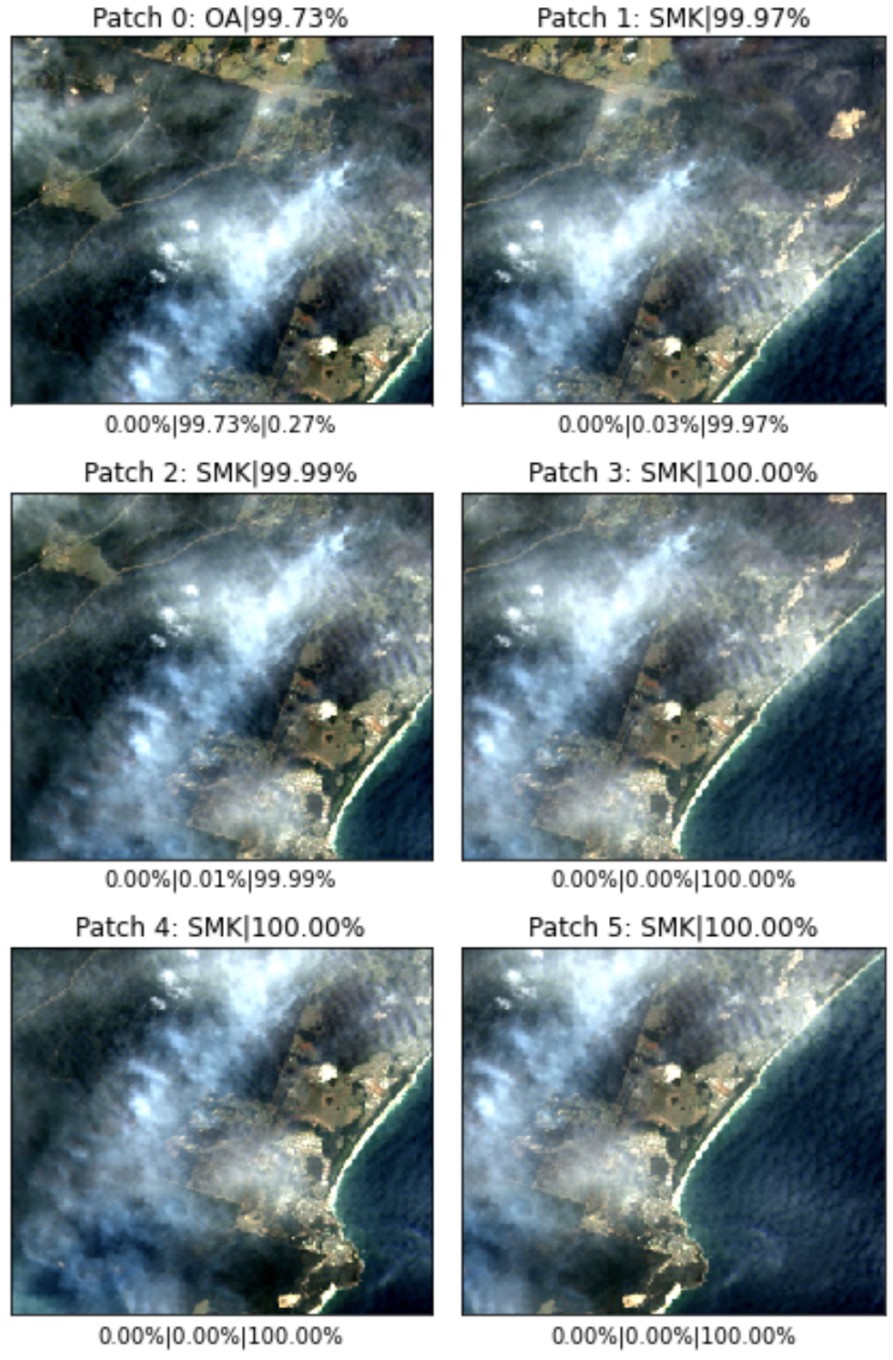

**Figure 10.** Prediction results of scene (a) in Figure 9.

In Figure 10, all patches were correctly predicted, except patch 0 was falsely predicted as "Other_aerosol".

The prediction results of scene (b) are shown in Figure 11.

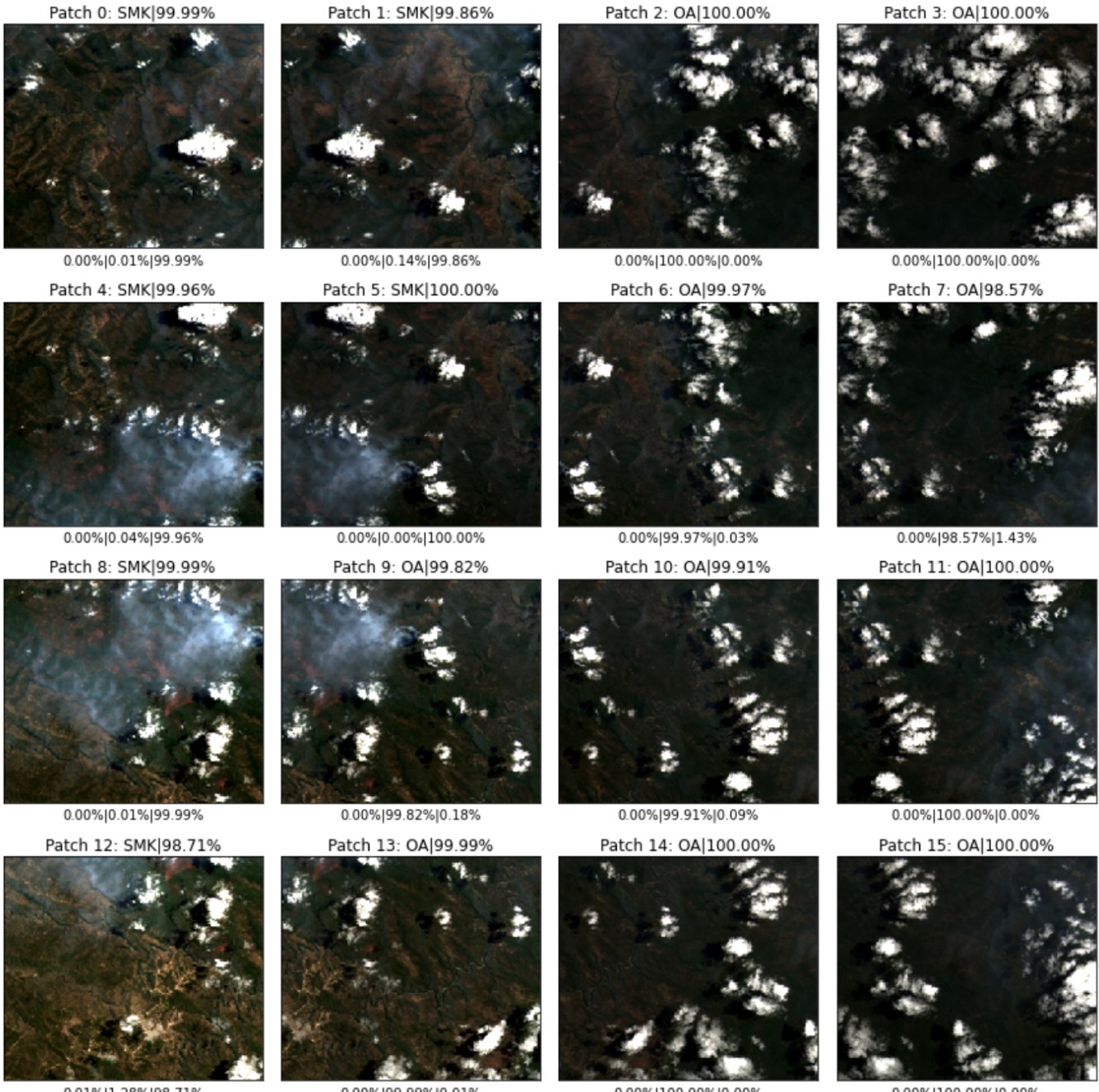

**Figure 11.** Prediction results of scene (b) in Figure 9.

In Figure 11, patches 0, 1, 4, 5, 8, 12 were correctly predicted as "Smoke". All other patches should also be "Smoke" but were falsely predicted as "Other_aerosol".

The prediction results of scene (c) are shown in Figure 12.

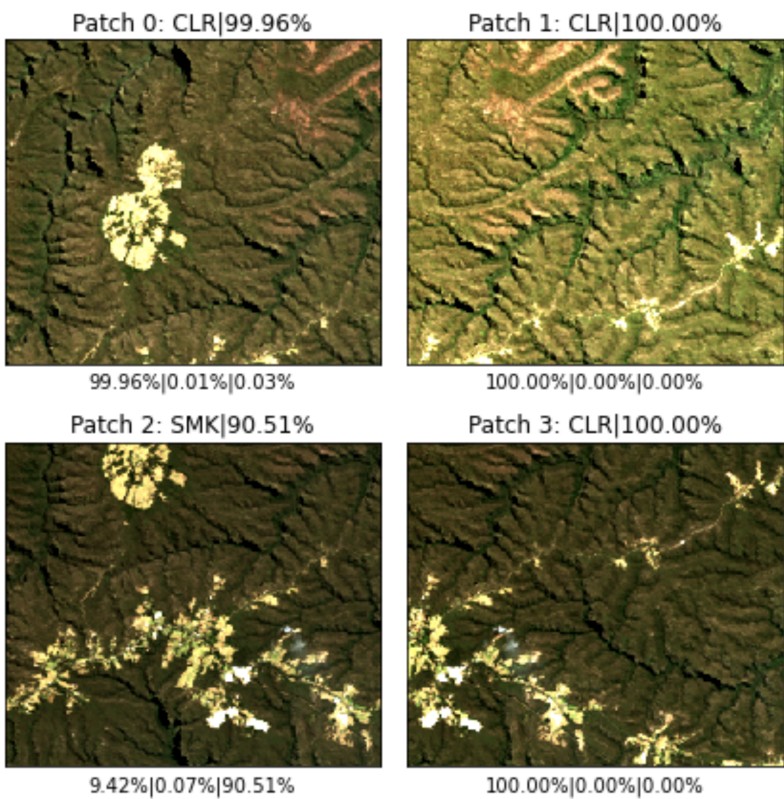

**Figure 12.** Prediction results of scene (c) in Figure 9.

In Figure 12, the two patches on the top were both correctly predicted as "Clear" with a high probability. The two patches on the bottom both contain fire smoke on a very small scale; however, only the left patch was correctly predicted; the right patch was falsely predicted as "Clear".

Since scene (d) is very large and showing the results of all patches is impractical, we selected the patches which have fire smoke in them to verify the prediction performance. The prediction results of the fire smoke area in the top right corner in scene (d) are shown in Figure 13. The prediction results of the fire smoke area in the middle of scene (d) are shown in Figure 14.

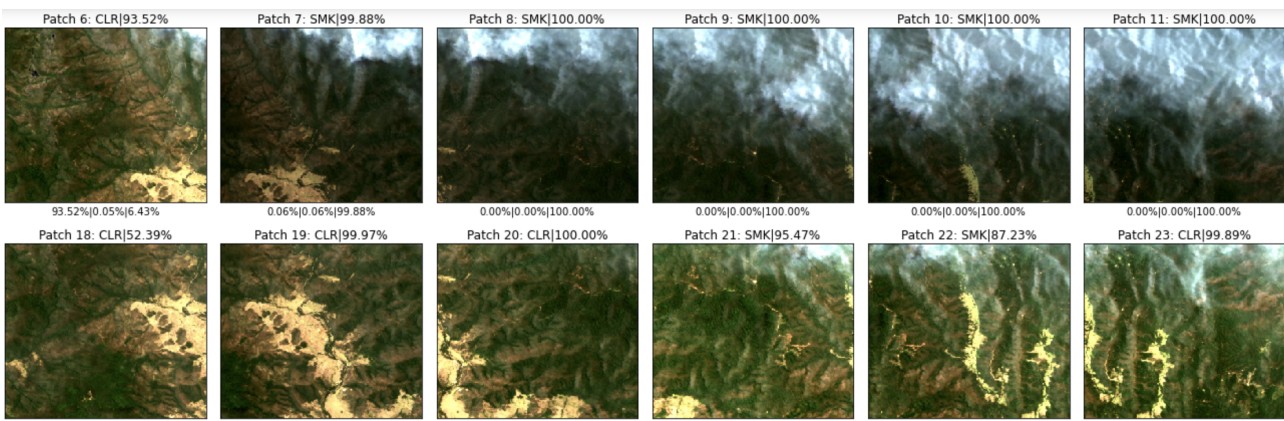

**Figure 13.** Prediction results of the top right fire smoke area of scene (d) in Figure 9.

In Figure 13, almost all the patches were correctly predicted, except patch 6 and patch 23, which both contain fire smoke but were falsely predicted as "Clear".

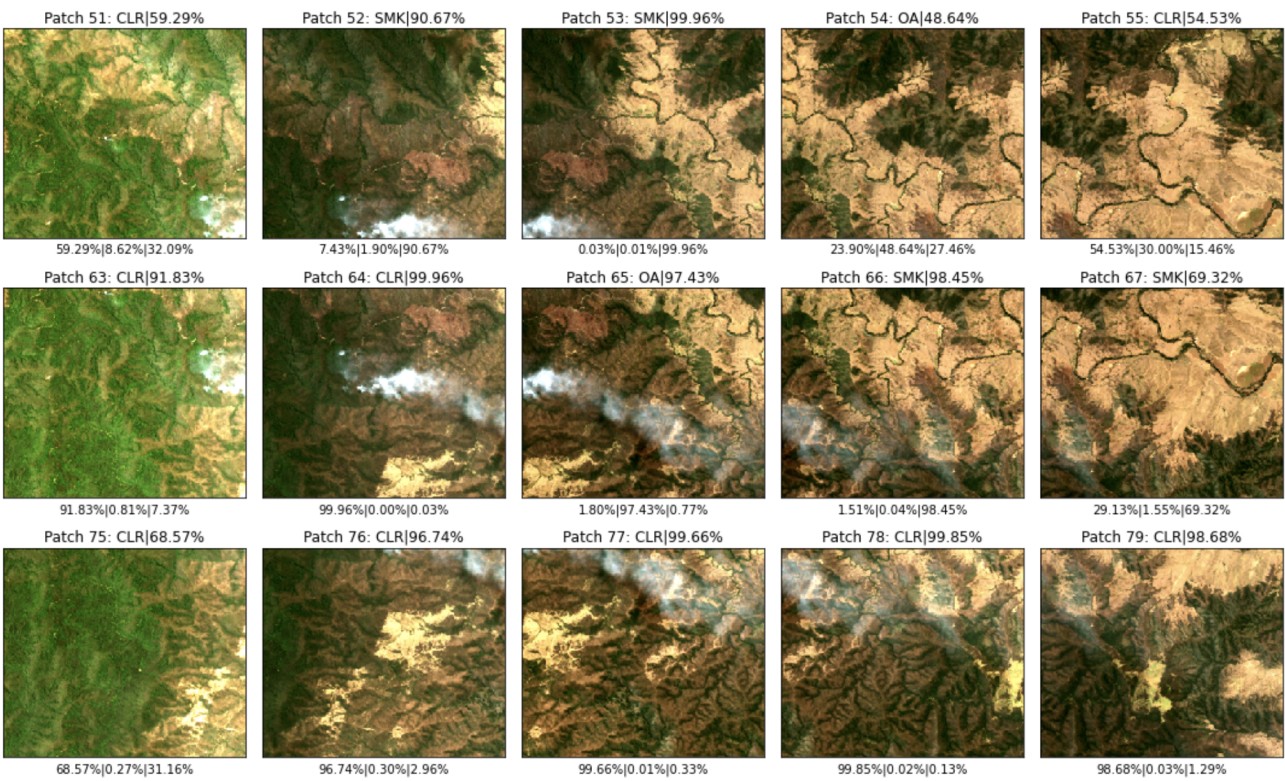

**Figure 14.** Prediction results of the middle fire smoke area of scene (d) in Figure 9.

In Figure 14, patches 52, 53, 66, and 67 were correctly predicted as "Smoke", whereas patches 51, 63, 64, 65, 76, 77, 78, and 79 were falsely predicted as either "Clear" or "Other_aerosol".

The above results demonstrate that:

- The VIB_SD_RGBNS1S2 model has a good overall accuracy;
- Fire smoke in the Landsat imagery can be detected, although sometimes with false negatives;
- False negative detection more likely happens where the fire smoke is small or mixed with clouds.

## 6. Discussion

The results shown in Table 9 complied with our expectation that using more spectral bands can effectively improve the model prediction accuracy. However, the results shown in Table 8 and Figure 8 posted one question: why did both VIB_SD_RGBNS2 and VIB_SD_RGBNS1N2 yield worse overall performance compared with VIB_SD_RGBN and VIB_SD_RGBNS1.

A few factors could be related to this problem. One factor may be linked to the imbalance of the fire smoke scenes in the dataset. This may be inferred from the skewed distribution of the accuracy and kappa-coefficient of the models in Figure 8. Fire smoke could be obscured by clouds, in dark colours hidden in a dark background, in the corners or on the edges of the images, or simply too small or too thin. When randomly splitting the samples, the training samples may majorly contain some of the fire smoke scenes, but the testing samples majorly contain other fire smoke scenes. This can lead to compromised training and testing performance. In contrast, when the testing samples comply with the training samples, the models would be likely to achieve better training and testing performance. In this case, the overall performance of the models might change if they are trained more times. However, repeating the training process is very time-consuming and will not guarantee to show us different results. We may need to expand the dataset in a

proper way to yield a more evenly distributed training and testing samples in the random splitting process.

Another factor could be related to whether the model can learn the information in the additional bands appropriately. The backpropagation process needs to effectively update the weights of all the input bands and extract useful features for the prediction. If the weights of some spectral bands cannot be properly updated, such bands would be treated as noises by the model and lead to counterproductive performance. This could be associated with the size of the training dataset or the model design. To solve this, we may need to train the models using a larger dataset or further adjust the model design. In our future work, we will look to expand the current dataset and fine-tune our model structure for further investigation.

The case study results in Section 5 demonstrate that the VIB_SD_RGBNS1N2 model can effectively detect fire smoke. Particularly, fire smoke mixed with or obscured by clouds in scene (a) and (b) and fire smoke over a very small geographic extent in scene (c) were correctly detected in one or more patches despite false negative predictions occurring in some other fire smoke patches.

Although the prediction stability of the model still needs to be improved, it has the potential to be used for early fire detection. We notice that while a few "Smoke" patches were misclassified as "Clear" or "Other_aerosol", "Clear" or "Other_aerosol" patches were rarely misclassified as "Smoke". This implies the false positive rate in predicting "Smoke" is low, so the positive prediction for "Smoke" should be trustworthy. Since the prediction is conducted on overlapped patches, although a few fire smoke patches might be misclassified, the fire alarm could be still triggered, provided one fire smoke patch can be correctly predicted.

Nonetheless, we will aim to improve the prediction stability in our future work. Apart from expanding the training data and adjusting the model design, we will examine whether the thermal band can further improve the model performance.

Furthermore, we will try to use multi-source satellite imagery to achieve timely detection of early fire smoke. The temporal resolution of Landsat 8 OLI is eight days, which can hardly satisfy the demand for timely detection. However, multiple low temporal resolution satellites will collectively provide a much higher temporal resolution. Therefore, we will also collect imagery datasets from more satellites (e.g., sentinel-2) in future research, aiming to timely detect early fire smoke using imagery from multiple satellites.

## 7. Conclusions

To facilitate satellite-based scene-level fire smoke detection, we constructed a multi-spectral imagery dataset from moderate spatial resolution satellites: Landsat 5 TM and Landsat 8 OLI. We developed a lightweight model structure VIB_SD that could be potentially adopted for on-board-of-small-satellite applications with significantly reduced parameters but only minor compromises in the accuracy. Based on VIB_SD, we trained five models with the dataset using different band combinations to evaluate the effectiveness of using multispectral moderate spatial resolution imagery in early fire smoke detection. Our experiment results demonstrated that training the models using all three additional IR bands can effectively improve the detection accuracy. We used the VIB_SD_RGBS1S2 model to conduct predictions on real fire smoke scenes. The results showed that the model can effectively detect early fire smoke in various scenarios, although the prediction stability still needs further investigation. Our future work will aim to refine the VIB_SD structure, expand the current dataset, collect new datasets from other satellites (e.g., Sentinel-2), and try to integrate multiple data sources for the timely detection of early fire smoke.

**Author Contributions:** Conceptualisation, L.Z., J.L. (Jixue Liu), S.P. and J.L. (Jiuyong Li); methodology, L.Z., J.L. (Jixue Liu), S.P. and J.L. (Jiuyong Li); software, L.Z.; validation, L.Z., J.L. (Jixue Liu), S.P., J.L. (Jiuyong Li), S.O. and N.M.; formal analysis, L.Z., J.L. (Jixue Liu) and J.L. (Jiuyong Li); investigation, L.Z.; resources, J.L. (Jixue Liu), S.P., S.O. and N.M.; data curation, L.Z.; writing—original draft preparation, L.Z.; writing—review and editing, L.Z., J.L. (Jixue Liu), S.P., J.L. (Jiuyong Li), S.O. and

N.M.; visualization, L.Z.; funding acquisition, J.L. (Jixue Liu) and J.L. (Jiuyong Li). All authors have read and agreed to the published version of the manuscript.

**Funding:** This work has been supported under the project P3-07s by the SmartSat CRC, whose activities are funded by the Australian Government's CRC Program.

**Data Availability Statement:** The Landsat imagery dataset used for this study is available upon request to the authors. The historical fire events datasets are hosted on Data NSW at https://data.nsw.gov.au/search/dataset/ds-nsw-ckan-1a66c7d9-a5eb-4d48-9c9f-8c804d65e1a1/details?q=fire (accessed on 25 June 2021) and DATA SA at https://data.sa.gov.au/data/dataset/fire-history (accessed on 25 June 2021). The USTC_SmokeRS dataset is available for download, the download links can be found on https://webpages.charlotte.edu/cchen62/dataset.html (accessed on 27 May 2021).

**Conflicts of Interest:** The authors declare no conflict of interest.

## Abbreviations

The following abbreviations are used in this manuscript:

| | |
|---|---|
| CNN | Convolutional Neural Network |
| DEA | Digital Earth Australia |
| DL | Deep Learning |
| FCN | Fully Convolutional Network |
| GIEP | Global Information Extraction Path |
| IR | Infra-Red |
| MAI | Mutual Activation Interim |
| MLP | Multi-Layer Perceptron |
| MODIS | Moderate Resolution Imaging Spectroradiometer |
| NIR | Near-Infra-Red |
| NSW | New-South-Wales |
| SA | South-Australia |
| SFEP | Salient Feature Extraction Path |
| SWIR | Short-Wave Infra-Red |
| VIB_SD | Variant Input Bands for Smoke Detection |

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
