# Peer review of "Investigating the Impact of Using IR Bands on Early Fire Smoke Detection from Landsat Imagery with a Lightweight CNN Model"

_remotesensing, doi:10.3390/rs14133047_

Round 1
Reviewer 1 Report
In this paper the authors have constructed a multispectral imagery dataset from Landsat 5 TM and Landsat 8 OLI. They have also proposed a DL model for fire smoke detection. Their experiment results demonstrated that the proposed model can effectively detect early fire smoke in various scenarios. In my opinion, this paper addresses an interesting topic and offers some innovation. The manuscript is definitely worth publishing.
My minor comments are attached.

Reviewer 2 Report
In view of the challenges faced by fire smoke detection today, this paper improves the accuracy of fire smoke detection by using high-resolution datasets and additional spectral bands. The reviewers believe that this work is not innovative enough, and the paper needs many revisions. The following points are also made:
1. The main contribution of the paper needs to be further expanded. Additional spectral bands can help identify more scene information for most other scenes, which is not enough to reflect the innovation of the paper.
2. The first two points of Section 3.3.1 need to be further expanded.
3. Section 3.3.2 needs to be adjusted, and the introduction of the loss function should be placed at the end of the model description.
4. The VIB_SD model proposed in this paper is not listed in the main contributions of the paper. So, whether the model is changed from the model proposed by others, and if so, the author should emphasize the changed part of the model.
5. The paper lacks ablation experiments.
6. The highlighting of the data in Table 9 is confusing.
7. There is an error in reference 17. Authors should carefully check all reference formats.
Reviewer 3 Report
1. I advise you to claim your novelty "Variant Input Bands for Smoke Detection" more firmly in the introduction, conclusion, and others
2. I suggest you modify the title to highlight your novelty.
Round 2
Reviewer 2 Report
My previous concerns have been modified. There is no more comments.